# MGMT and Whole-Genome DNA Methylation Impacts on Diagnosis, Prognosis and Therapy of Glioblastoma Multiforme

**DOI:** 10.3390/ijms23137148

**Published:** 2022-06-27

**Authors:** Rosa Della Monica, Mariella Cuomo, Michela Buonaiuto, Davide Costabile, Raduan Ahmed Franca, Marialaura Del Basso De Caro, Giuseppe Catapano, Lorenzo Chiariotti, Roberta Visconti

**Affiliations:** 1CEINGE-Advanced Biotechnologies, Via G. Salvatore 486, 80145 Napoli, Italy; dellamonica@ceinge.unina.it (R.D.M.); mariella.cuomo@unina.it (M.C.); buonaiutom@ceinge.unina.it (M.B.); costabile@ceinge.unina.it (D.C.); 2Department of Molecular Medicine and Medical Biotechnologies, University of Napoli “Federico II”, Via S. Pansini 5, 80131 Napoli, Italy; 3SEMM-European School of Molecular Medicine, University of Napoli “Federico II”, Via G. Salvatore 486, 80145 Napoli, Italy; 4Department of Pathology, University of Napoli “Federico II”, Via S. Pansini 5, 80131 Napoli, Italy; aduanahmed.franca@unina.it (R.A.F.); marialaura.delbasso@unina.it (M.D.B.D.C.); 5Neurosurgery Unit, “Ospedale del Mare” Hospital, Via E. Russo 11, 80147 Napoli, Italy; giuseppecatapano@libero.it; 6Institute for the Experimental Endocrinology and Oncology “G. Salvatore”, National Council of Research of Italy, Via S. Pansini 5, 80131 Napoli, Italy

**Keywords:** glioblastoma multiforme, *MGMT* gene methylation, whole-genome methylation profiling, liquid biopsy, IDH1/2 mutations

## Abstract

Epigenetic changes in DNA methylation contribute to the development of many diseases, including cancer. In glioblastoma multiforme, the most prevalent primary brain cancer and an incurable tumor with a median survival time of 15 months, a single epigenetic modification, the methylation of the *O^6^-Methylguanine-DNA Methyltransferase* (*MGMT*) gene, is a valid biomarker for predicting response to therapy with alkylating agents and also, independently, prognosis. More recently, the progress from single gene to whole-genome analysis of DNA methylation has allowed a better subclassification of glioblastomas. Here, we review the clinically relevant information that can be obtained by studying *MGMT* gene and whole-genome DNA methylation changes in glioblastomas, also highlighting benefits, including those of liquid biopsy, and pitfalls of the different detection methods. Finally, we discuss how changes in DNA methylation, especially in glioblastomas bearing mutations in the *Isocitrate Dehydrogenase (IDH) 1 and 2* genes, can be exploited as targets for tailoring therapy.

## 1. Introduction

Among mammalian organs, the brain contains the highest levels of DNA methylation [1]. The majority of DNA methylation consists of the transfer of a methyl group on cytosines preceding guanines on CpG sites to form 5-methylcitosine. However, in mammalian neurons, cytosines are frequently methylated also when followed by any other nucleotide than G [2]. Generally speaking, DNA methylation influences gene expression, with its effects depending on where it is located (regulatory regions, gene bodies, intergenic regions). In addition to its functional effects, DNA methylation can be utilized as a marker for diagnosis and prognosis of many common cancers, including brain tumors [3]. Here, we review how DNA methylation of a single gene (i.e., *the O^6^-Methylguanine-DNA Methyltransferase, MGMT, gene*) as well as the entire genome (i.e., the methylome) is currently utilized for refining glioblastoma diagnosis, prognosis and treatment. Finally, we highlight how aberrant DNA methylation is also pursued as a novel target for glioblastoma therapy.

## 2. Relevance of *MGMT* Methylation Assessment for Glioblastoma Clinical Management

In brain tumors, the methylation of the regulatory regions of one gene, *MGMT*, is actively investigated for its significance as a prognostic biomarker, mainly because it predicts response to temozolomide treatment [4].

Temozolomide is an alkylating drug approved for the treatment of adult patients with newly diagnosed glioblastoma as an adjuvant to surgery (if surgery is feasible) and concomitantly with radiotherapy. After, temozolomide is the drug of choice for the maintenance regimen. Certainly, currently, the most effective chemotherapy agent for patients with gliomas, temozolomide is well-tolerated, most of the toxicities being of low grade [5]. Commonly, temozolomide causes thrombocytopenia and gastro-intestinal side effects, such as nausea, loss of appetite and vomiting [5,6]. To further improve temozolomide efficacy, new combinatorial regimens are eagerly tested. Recently, it has been reported that nanoparticles decorated with bromodomain inhibitors enhance temozolomide efficacy in glioma therapy [7]. Moreover, the results of recent, promising preclinical studies suggest that temozolomide efficacy might be improved by targeting DNA polymerase eta or by inhibiting the NFAT1/FasL signaling, opening novel therapeutic possibilities for glioblastoma treatment [8,9].

As shown in Figure 1, temozolomide functions because its metabolites cause methylation of the DNA purine bases in O^6^-guanine, N^7^-guanine and N^3^-adenine, O^6^-methylguanine (O^6^-MeG) being the principal cytotoxic lesion, although accounting for only about 5% of the total adducts formed by the drug [10]. During DNA replication, O^6^-MeG can be recognized as adenine, inducing mismatched incorporations of thymine instead of cytosine. While the mismatched thymine is efficiently removed by the DNA mismatch repair pathway, the O^6^-MeG stands, resulting in futile cycles of repair leading to DNA nicks and, in turn, to cell-cycle arrest and/or apoptosis. 

That the 0^6^-MeG temozolomide-induced adduct is the most prominent lesion is indirectly demonstrated by the evidence that glioblastomas with the reduced or absent activity of the MGMT enzyme are more sensitive to the drug. MGMT removes methyl adducts from the O^6^ position of guanine, thus antagonizing temozolomide effects. MGMT is a “suicide” enzyme: during the repair process, the methyl moiety is transferred from the O^6^-methylguanine to the MGMT protein itself, resulting in its irreversible inhibition.

Initial studies demonstrated that in MGMT activity null- or deficient-cells, the 5′ CpG islands in the *MGMT* gene were highly methylated, indeed causing silencing of the gene [11,12,13]. Later, following promising preliminary clinical evidence [13,14], a definitive clinical trial has demonstrated that only patients with a methylated *MGMT* gene benefited from temozolomide addition to radiotherapy, being their median survival significantly longer [15]. In detail, the median survival of patients treated with temozolomide and radiotherapy was 21.7 months (95 percent confidence interval, from 17.4 to 30.4), as compared with the median survival of 15.3 months (95 percent confidence interval, from 13.0 to 20.9) of patients assigned to only radiotherapy (*p* = 0.007 by the log-rank test). By contrast, among patients whose tumors were not *MGMT* methylated, the addition of temozolomide to radiotherapy had only a marginally significant impact on survival; this trial finally established *MGMT* gene methylation status as a biomarker for predicting response to temozolomide and, in turn, a powerful prognostic factor [15]. Further on, in a similar patient population, with newly diagnosed glioblastoma and treated with radiotherapy plus temozolomide, *MGMT* methylation status has been shown to be strictly correlated to timing and pattern of recurrence. *MGMT*-silenced, temozolomide-treated patients suffered recurrence after a longer time interval (14.9 versus 9.2 months, *p* = 0.02) and outside the radiation field, enforcing the role of *MGMT* gene methylation in glioblastoma prognosis [16]. Overall, the results of these clinical studies have led to the recommendation of investigating *MGMT* methylation status in all patients with newly diagnosed glioblastomas. More recently, a phase III multicentric trial has shown that temozolomide, in addition to short-course radiation therapy, doubled median survival in elderly (>65 years) glioblastoma patients with methylated *MGMT* gene with respect to patients with unmethylated *MGMT* gene (13.5 months versus 7.7 months, the hazard ratio for death, 0.53; 95% CI, from 0.38 to 0.73; *p* < 0.001) without increasing toxic effects. The addition of temozolomide to short-course radiotherapy had slighter but still significant benefits also in patients 65 years of age or older with unmethylated *MGMT* (10.0 months versus 7.9 months, the hazard ratio for death, 0.75; 95% CI, from 0.56 to 1.01; *p* = 0.055), again reinforcing the concept that *MGMT* methylation status is, at least in elderly patients, a valid biomarker for predicting not only response to temozolomide but also, independently, prognosis [17].

Investigation of the *MGMT* gene methylation status can also help in discriminating recurrence from pseudoprogression. Glioblastoma pseudoprogression is an MRI image mimicking tumor progression; it is, however, transient: it occurs typically during the first three months after therapy and improves quite quickly, often within a few weeks. Pseudoprogression occurs in up to 90% of patients with *MGMT* gene methylation upon treatment with temozolomide and radiotherapy [18,19]. On the other hand, pseudoprogression is quite rare in patients with unmethylated *MGMT*, with the pseudoprogression MRI pattern being a consequence of the necrosis caused by effective temozolomide action [20]. 

Overall, knowledge of *MGMT* methylation status is critical for clinicians to better personalize the care of glioblastoma patients.

## 3. Techniques for DNA Methylation Testing in Glioblastomas

### 3.1. Techniques for MGMT Methylation Assessment in Glioblastomas

Methylation-specific PCR (MSP) is the golden standard for *MGMT* methylation assessment in glioblastomas [21]. 

MSP can rapidly assess the methylation status of any CpG site within a CpG island [22,23]. In a first step, DNA is treated with sodium bisulfite, resulting in the conversion of unmethylated cytosine into uracil while the methylated cytosine, resistant to bisulfite, remains unaltered. Thus, MSP, as most of the techniques shown in Figure 2, utilized for investigating *MGMT* methylation status, requires bisulfite conversion of DNA. 

Polymerase chain reaction amplification is then performed with two sets of primers designed to anneal to methylated cytosines or to the bisulfite-modified, unmethylated cytosines. Methylation status at a CpG site is determined by which specific primer set achieves DNA amplification. *MGMT* methylation status is routinely investigated in glioblastoma utilizing primers, decisively validated in clinical trials, interrogating, within exon 1, 9 CpG sites [13,24]. PCR products are, finally, run on agarose gels with appropriate negative and positive controls and samples approximately equivalent to the positive methylated control are called methylated. Thus, MSP is a qualitative technique, the results of which are based on the presence/absence of methylation in the regions where primers anneal. The technique does not allow the identification of specific methylated cytosines or even just quantification of the exact number of methylated CpG sites. Even more problematic, from a diagnostic point of view, it remains very difficult to determine validated, and standardized among different laboratories, cut-offs for calling *MGMT* “methylated” or “unmethylated”. However, mostly because of its simplicity and low cost, MSP is still the best method for *MGMT* methylation assessment in glioblastomas. 

The more recently developed techniques are mostly utilized for research purposes or have been used in selected clinical trials. As an example, a real-time, methylation-specific PCR assay has been utilized in a trial designed to correlate, in newly diagnosed glioblastoma patients, the responses to dose-dense temozolomide with *MGMT* methylation [25]. Additionally, in this case, specificity was obtained by selective amplification of bisulfite-modified DNA, but a semiquantitative result was achieved by normalizing the number of copies of methylated *MGMT* to the number of copies of a housekeeping gene [26]. 

*MGMT* methylation has also been quantified through a high-resolution melt analysis, comparing differences in melting of unmethylated and methylated sequences to standards with known unmethylated to methylated ratio [27]. Although successful in predicting glioblastoma response to temozolomide, the methylation-sensitive high-resolution melt has not been validated for investigating *MGMT* methylation in clinical settings, thus still lacking cut-off thresholds [28].

The only quantitative technique standardized across laboratories for investigating *MGMT* methylation is pyrosequencing. Pyrosequencing, a “sequencing-by-synthesis” method, allows the sequencing of a single strand of DNA by synthesizing the complementary strand and detecting which base is incorporated at each step by a DNA polymerase. When used with bisulfite conversion and quantitative PCR, it can effectively determine the methylation status of each *MGMT* CpG dinucleotides. The concordance between MSP and pyrosequencing has been reported to be high, at least in assessing *MGMT* methylation in glioblastomas [29]. Accordingly, a recent prospective multicenter trial ended up showing that semiquantitative MSP and pyrosequencing are both successful in evaluating the predictive value of *MGMT* methylation in survival, pyrosequencing having, in addition, the greater reproducibility among the participating clinical centers. Conclusions partially disproven by three comparative studies finding that *MGMT* methylation status assessed by pyrosequencing is, indeed, more reliable for predicting the survival of glioblastoma patients [30,31,32]. In waiting for more conclusive studies, pyrosequencing remains, today, largely too costly to be utilized routinely in clinical settings.

All the above-mentioned techniques require DNA treatment with sodium bisulfite. In order to skip this time-consuming and often poorly efficient step, methylation-specific multiplexed ligation-dependent probe amplification (MS-MLPA) has been explored as a method for investigating *MGMT* methylation. MS-MLPA, the gold standard technique for studying methylation of imprinted genes in patients with suspected imprinting disorders, is based on the use of the restriction endonuclease HhaI, sensitive to methylation in its GCGC restriction site. In MS-MLPA, if the CpG locus is not methylated, the enzyme cleaves the restriction site, resulting in a lack of PCR amplification; if the CpG locus is methylated, the HhaI restriction site is not digested, resulting in the generation of a PCR product. In comparative studies, MS-MLPA has given information on *MGMT* methylation status concordant with that obtained with other techniques, including MSP [31,33,34,35]. Despite other noteworthy features, such as the capacity to give semiquantitative results, MS-MLPA is not, however, the standard diagnostic method for *MGMT* methylation status screening, this being to date, as stated above, MSP.

### 3.2. Whole-Genome Methylation Profiling (Methylome) of Glioblastomas 

The whole-genome DNA methylation profile (methylome) of a tumor is the result of both somatically acquired changes and of features reflecting the cell of origin [36]. Thus, methylome analysis has been successful in both subclassifying tumors previously considered homogeneous diseases and in tracing the origin of undifferentiated metastases of cancers of unknown primary [37,38,39].

As shown in Figure 3, the entire epigenomic tumor profile can be investigated using different genome-wide, high-throughput platforms, such as the Illumina Infinium HumanMethylation450 BeadChip (450 k) or the newest Methylation BeadChip (EPIC) array, covering 850,000 CpG sites. 

In a pivotal paper, the re-classification of CNS tumors on the basis of their methylation signature resulted in a change of diagnosis in up to 12% of the cases, demonstrating how informative the methylome is [40]. Moreover, using ad hoc bioinformatic pipelines, information can be extrapolated, by the whole-genome data, about the methylation status of every single gene, including *MGMT*, and about gene “copy number variation” (CNV). By CNV analysis, large chromosomal rearrangements and loss or acquisition of material for single genes and/or entire chromosomes can be readily recognized. More narrowing bioinformatic pipelines can be used to also assess the co-deletion 1p-19q, a prerequisite for oligodendroglioma diagnosis, or the therapeutically targetable Epidermal Growth Factor Receptor (EGFR) amplifications. 

Methylome profiling of glioblastoma is costly and technically challenging, and its reliability also depends on the percentage of tumor cells present in the analyzed tissue. On the one hand, thus, the quantitative measurement of DNA methylation by genome-wide, high-throughput platforms is not, at the moment, affordable for all the clinical centers; on the other hand, it is a plus that some centers, including ours, do use in clinical practice to give more, useful information for glioblastoma management [21,41]. 

To improve the whole-genome DNA methylation-based classification of glioblastomas, the pitfall of their highly heterogenicity has to be further dealt with. In glioblastoma, tumor cells with different characteristics have been identified, recapitulating neural development and, thus, named, on the basis of their major genotypic and phenotypic features, as neural-progenitor-like, oligodendrocyte-progenitor-like, astrocyte-like and mesenchymal-like cells [42,43]. Each glioblastoma comprises cells belonging to the four types, although at different frequencies. Adding complication, during tumorigenesis, plasticity has been demonstrated between tumor cell types, modulated by genetic drivers and influenced by the tumor microenvironment [43]. Thus, glioblastomas are highly heterogenous, both at the molecular and at the cellular level, and there is high variability both within and between tumors. 

Additionally, DNA methylation is highly variable in glioblastomas, with, remarkably, some samples exhibiting higher differences in DNA methylation within tumors than between tumors. Moreover, within a tumor, a high percentage of CpG sites have different methylation levels [44]. Such a high variability has to be borne in mind when DNA methylation is used for tumor classification and subtyping. Numerous approaches are under investigation to tackle the problem, including, intuitively, the analysis of more than one biopsy taken from different areas within the tumor mass or the more complex studies on single tumor cells, expanded in culture as single-cell clones [45,46]. Still, to date, glioblastoma intratumor heterogeneity negatively impacts the methylome profile-based classification, and more research is needed to further improve the usefulness of these molecular investigations for clinicians. Moreover, whole-genome analyses also need improvement to address the role of non-CpG methylation in the development and progression of cancer, including glioblastoma [47]. Parenthetically, glioblastoma heterogenicity can also affect the prognostic/predictive value of the *MGMT* gene methylation status [48]. To improve, a novel analysis, considering separate cut-off values for calling each individual CpG as methylated or unmethylated in the *MGMT* gene, has been proposed [49,50].

### 3.3. DNA Methylation Analysis of Glioblastomas by Nanopore, “Third-Generation” Sequencing

Nanopore sequencing, as shown in Figure 4, is a new technology that works by registering changes to an electrical current as nucleic acids, DNA or RNA pass through a protein nanopore [51]. The resulting signal is directly decoded to provide the specific sequence.

Thus, nanopore sequencers allow real-time analysis of DNA or RNA fragments and, importantly, provide the longest read lengths, up to 2 Mb. Moreover, the library preparation is relatively easy and of great relevance for DNA methylation studies; nanopore sequencers can detect 5-methylcytosine modifications in native DNA without the need for the time-consuming and often inefficient bisulfite conversion. Even though the smaller, more handy devices, such as the portable MinION, yield low-coverage sequences, first evidence has been published showing how nanopore technology, if well implemented, can be utilized for brain tumor characterization. A pilot study on 45 glioma samples has demonstrated that nanopore sequencing allowed to classify brain tumors on the basis of the whole-genome DNA methylation profile with precision comparable to the EPIC array [52]. Moreover, nanopore detected *MGMT* promoter methylation with the same accuracy as pyrosequencing and EPIC array in all the investigated cases [52]. Thus, in the near future, nanopore sequencing might indeed be a lower cost and less time-consuming alternative method for *MGMT* gene methylation assessment and for methylome-based classification of glioblastomas.

Nanopore potentials have been further improved by the nanopore Cas9-targeted sequencing (nCATS) strategy. nCATS uses Cas9 to specifically target and cut chromosomal DNA, then ligate adapters for nanopore sequencing. nCATS can simultaneously assess single-nucleotide variants, structural variations and CpG methylation [53]. Accordingly, nCATS has been successful in simultaneously detecting methylation of the *MGMT* gene and mutations in *Isocitrate Dehydrogenase (IDH) 1/2* genes, these are necessary, as discussed below, for better narrowing glioblastoma diagnosis [54].

The nanopore’s peculiar ability to generate sequences in real-time and, moreover, without the need of the time-consuming bisulfite DNA conversion procedures, opens even more opportunities for better managing brain tumors. Thus, nanopore has been successfully used to obtain intraoperatively whole-genome methylation profiles of brain tumors, including glioblastomas [55]. Intraoperative nanopore sequencing combined with machine learning diagnostics has allowed tumor classification, concordant with those obtained upon a complete, standard neuropathological evaluation, in 89% of the cases [55]. Importantly, the results were returned to the neurosurgeon at a median of 97 min [55]. Knowing intraoperatively precise tumor typing is key for the surgeon to decide the better surgical strategy, a great help in choosing between maximal resection, whose benefits depend, indeed, on the tumor characteristic, and the risk of severe brain damage. Strikingly, an intraoperative diagnosis of a low-grade glial-neuronal tumor can even lead to the decision that cytoreduction is not indicated. On the other hand, multiple lesions, intraoperatively diagnosed as multifocal diffuse gliomas, can stop the surgeon, as the risks of radical resections outweigh the expected benefits.

Besides the more futuristic applications such as the intraoperative use, overall, it is predictable that nanopore sequencing, thanks to its peculiar ability to provide, more quickly and effortlessly, information on gene mutation and methylation, will be soon a broadly used alternative to more conventional sequencing methods.

## 4. Glioblastoma DNA Methylation Assessment in Liquid Biopsies

Liquid biopsies are a further improvement in the present era of precision medicine. Circulating tumor cells, cell-free DNA and vesicles such as exosomes, isolated from blood or from other body fluids, including the cerebrospinal fluid, have been used to obtain information about cancer patients. First, the number of circulating tumor cells has been instrumental in predicting prognosis [56]. Then, technical improvements in the isolation of circulating cancer material and in sequencing have made it possible to obtain from liquid biopsies information about genetic and epigenetic alterations of solid tumors, with broad implications on diagnosis and prediction of response to treatments. Although not already in clinical routine, liquid biopsies are expected to become in a few years an essential tool because they are non-invasive, repeatable and informative for inaccessible tumors [57]. Given these characteristics, liquid biopsies could be particularly useful for the diagnosis and the follow-up of glioblastoma patients and are, therefore, the object of study. 

*MGMT* gene methylation has been investigated in circulating cell-free DNA isolated from the serum of glioblastoma patients. When compared with the tumor tissue, the specificity of the analysis was 100%, but the sensitivity was as low as 51%, probably because the blood-brain barrier negatively affects the amount of circulating material in glioblastoma patients [58]. Accordingly, *MGMT* gene methylation can be more successfully detected in circulating DNA isolated from the cerebrospinal fluid, this approach being, however, not always feasible as the lumbar puncture is contraindicated in patients with increased intracranial pressure [59,60].

Interestingly, a recent study has shown that the DNA isolated from plasma of glioblastoma patients has lower levels of methylation on the Alu sequences than the healthy controls, Alu methylation negatively correlating with disease severity. Thus, if further proven, the methylation level of Alu in circulating DNA could be envisioned as an easily measurable prognostic biomarker [61].

Plasma cell-free DNA has also been utilized for genome-wide methylation profiling of brain tumors. Cell-free methylated DNA immunoprecipitation and high-throughput sequencing allowed highly sensitive discrimination of glioma patients from healthy controls or from patients with brain metastasis. Even more importantly, circulating methylome signature accurately discriminated different primary brain tumors, otherwise indistinguishable, using imaging techniques, non-invasively providing key information for planning the best therapeutical strategy [62]. However, still, the low quality and amount of plasma cell-free DNA (the methylated fraction being even smaller) remain major limitations for its use in methylome analyses.

Additionally, the DNA isolated from extracellular vesicles of glioblastoma patients allowed molecular classification of the tumors, with the same accuracy obtained by the methylome profiling of the tumor tissue [63]. Promisingly, the first study on a small cohort of 43 glioblastoma patients has shown that extracellular vesicle concentration is specifically higher in glioblastoma patients compared with healthy controls or with patients with brain metastases [64].

Overall, these studies encourage the pursuit of the methylome analysis of circulating DNA as a mean of monitoring glioblastomas.

## 5. *IDH1/2* Mutations and the Methylator Phenotype of Glioblastomas: New Therapeutic Targets

Mutations in the *Isocitrate Dehydrogenase (IDH) 1 and 2* genes are early driver genetic events during glial tumor development [65]. Accordingly, *IDH1/2* mutations are a hallmark of lower-grade gliomas, up to 90% of WHO grade II and up to 70% of WHO grade III gliomas harboring mutations in these genes [66]. Moreover, nearly all 1p-19q codeleted oligodendrogliomas bear an *IDH1/2* mutation, along with *Telomerase Reverse Transcriptase (TERT)* promoter mutation and *MGMT* gene methylation [65]. Finally, mutations in the *IDH1/2* genes are frequently found in a distinct subtype of glioblastomas, characterized by the younger age of onset and longer survival. These tumors, which evolved from low-grade gliomas, are identified as secondary glioblastomas and are characterized by a DNA hypermethylation signature; hence, the classification as Glioma CpG Island Methylator Phenotype: G-CIMP [67]. IDH mutant proteins have, in fact, severe metabolic impact, resulting in the competitive inhibition of many α-ketoglutarate-depending enzymes, including the DNA demethylase TET2 [68]. Deficiency in TET2 activity alters the DNA methylation/demethylation balance, resulting in the hypermethylation of several genes. Importantly, among the 263 genes found significantly downregulated and hypermethylated within G-CIMP tumors, many were associated, upon gene ontology analysis, with tumor invasion [69]. Indeed, IDH1/2 mutations have been established as the most powerful positive prognostic factor for glioma patient survival, followed by age, tumor grade and *MGMT* gene methylation status [65]. Moreover, typically, G-CIMP tumors have a distinct profile of copy number variation when compared to non-G-CIMP tumors [69].

Various therapeutic approaches have been tested in preclinical models to specifically target G-CIMP tumors, including the use of DNA methyltransferase inhibitors and IDH inhibitors [70,71,72]. The DNA methyltransferase inhibitor 5-azacytidine has been shown to reduce methylation, promote differentiation and, finally, induce tumor regression in a patient-derived IDH1 mutant glioma xenograft animal model [70]. Two IDH inhibitors, Enasidenib and Ivosidenib, have been approved by the FDA for the treatment of patients with refractory or relapsed acute myeloid leukemia (AML) bearing IDH2 mutations [73]. Promisingly, a novel, selective IDH1 inhibitor, Olutasidenib, used as a single agent or in combination with 5-azacytidine, has induced a deep clinical response in IDH1 mutated AML patients in phase I clinical trial [74]. A phase 1b/2 clinical trial (NCT03684811), investigating the effect of the 5-azacytidine in combination with Olutasidenib on glioblastoma patients with IDH1 mutations is ongoing, and the results are eagerly awaited [75]. 

## 6. Conclusions and Future Perspectives

Studying epigenetic changes in DNA methylation has provided important clinical information on glioblastomas. 

*MGMT* gene methylation status is a valid biomarker for predicting prognosis and response to therapy with alkylating drugs in glioblastomas. However, methods for detection are highly variable between laboratories, and critically, optimal cut-off definitions for *MGMT* status determination are still lacking. Thus, routine implementation in clinical practice is yet a challenge. 

Whole-genome methylation analysis of glioblastomas has allowed a narrower classification of tumor subtypes on the basis of the different methylome profiles. The epigenetic classification of gliomas has been proven to provide prognostic value independently of other known overall survival predictors such as age and grade [76]; however, also because of the important intra-tumoral heterogenicity of glioblastomas, further studies are still needed to definitively identify subtypes truly predictive of response to therapy and survival [77].

Assessing DNA methylation of both the *MGMT* gene and the whole genome in liquid biopsy is a key step in the path to improving personalized care of patients affected by not readily accessible glioblastoma tumors. In glioblastomas, liquid biopsies can be challenging because the blood–brain barrier could impede the release of tumor materials into the blood; however, on the other hand, the integrity of the barrier may be compromised, especially in advanced glioblastomas [78]. To date, glioblastoma DNA, both free circulating and enclosed in extracellular vesicles, has been successfully isolated and utilized for methylation analyses, encouraging further studies. Additionally, standardization of detection techniques, prospective studies for large-scale validation and more cost-effective methods are needed before transferring liquid biopsies into clinical practice.

Recent, important technological improvements, such as the nanopore, “third-generation” sequencing, pave the way for novel possibilities of utilizing DNA methylation for better glioblastoma clinical management. Above all, the intraoperative whole-genome methylation profile of brain tumors can be a game-changer for neurosurgeons and is surely worth more effort so that it can be routinely used as soon as possible.

The still relatively low throughput of the “third-generation-sequencing” methods makes DNA optical mapping an attractive alternative for studying DNA methylation signatures over large genomic fragments at single-molecule resolution. DNA optical mapping, recently successfully utilized for the genetic/epigenetic diagnosis of the facioscapulohumeral muscular dystrophy, precisely because providing long-read methylation profiles of single DNA molecules, can be instrumental for studying the methylation profiles of genes together with remote regulators such as distant enhancers [79,80]. Still far from being used routinely in clinical practice, the information that can be obtained by DNA optical mapping is a great example of what the future holds for glioblastoma treatment. Once established the relevance that glioblastoma DNA methylation of the single gene *MGMT* and of the whole genome has in narrowing classification and in indicating therapeutic approaches, all the progress in the techniques for detecting methylation will be of importance for glioblastoma cure, especially if, as DNA optical mapping, able to provide quantitative measurements.

A DNA hypermethylation signature characterizes glioblastomas harboring mutation in the *IDH* genes. Several preclinical studies have shown that IDH-mutated glioblastomas can be specifically targeted by hypomethylating drugs, such as 5-azacytidine. Clinical trials with such drugs, also in association with specific IDH inhibitors, are ongoing in glioblastoma patients after promising results in AML patients. The results are eagerly awaited by clinicians and patients struggling with glioblastoma, a tumor whose diagnosis and prognosis have been improved by the information also provided by DNA methylation studies, but that still remains largely incurable.

## Figures and Tables

**Figure 1 ijms-23-07148-f001:**
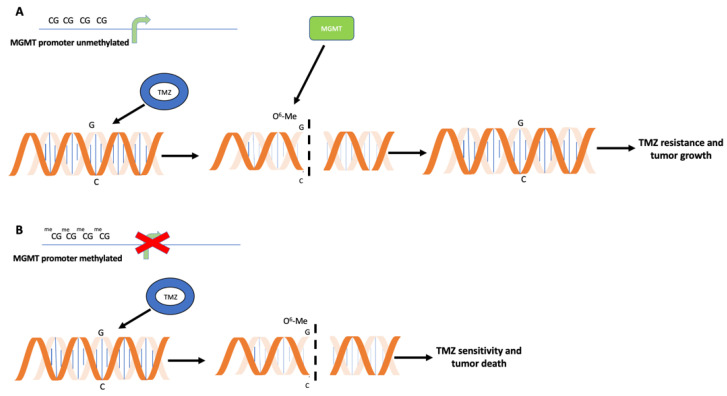
Antagonistic action between temozolomide (TMZ) and O^6^-Methylguanine-DNA Methyltransferase (MGMT) enzymatic activity. (**A**) TMZ drug (blue circle) modifies guanine (G) in O^6^-methylguanine (O^6^-MeG), creating mismatches in DNA sequence and, in turn, DNA breaks. Unmethylated *MGMT* promoter is associated with *MGMT* gene expression and MGMT protein synthesis (green rectangle). MGMT removes methyl-group on guanosine, re-establishing correct sequence. This event is associated with TMZ resistance, tumor growth and poor prognosis. (**B**) When *MGMT* promoter is methylated, the absence of *MGMT* transcription and synthesis results in TMZ tumor sensitivity, with accumulation of DNA mismatches and, in turn, cell death.

**Figure 2 ijms-23-07148-f002:**
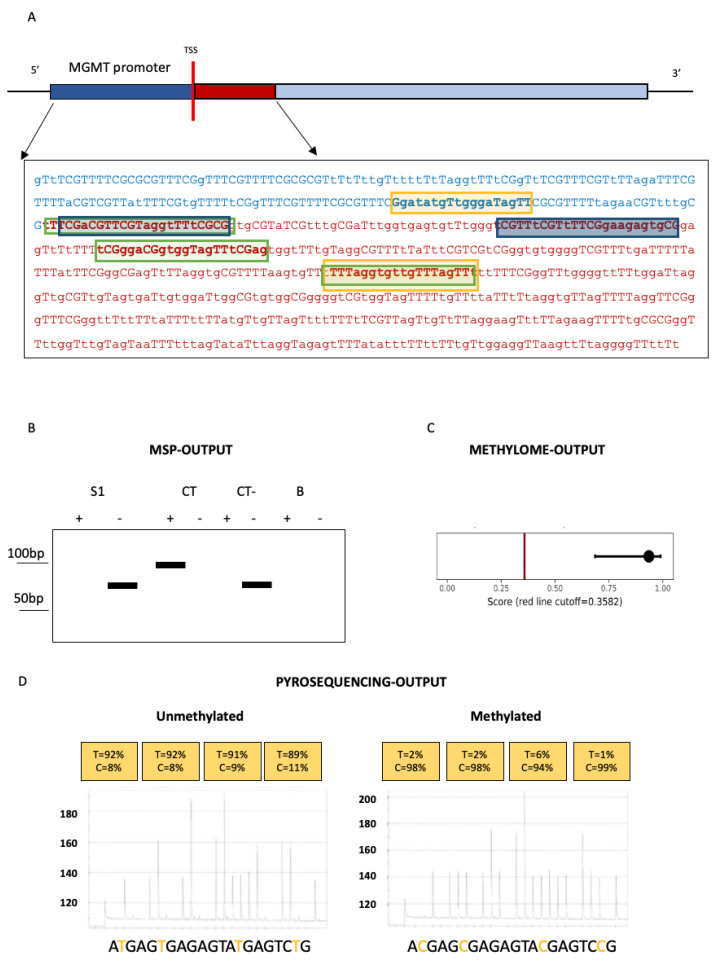
Graphic representation of *MGMT* gene regulatory region and output of the technologies used to analyze *MGMT* methylation after sodium bisulfite conversion. (**A**) *MGMT* gene promoter (blue) and exon 1 (red) are represented. Focus on bisulfite-converted sequence analyzed by different techniques: in yellow rectangles, primers used to perform pyrosequencing; in blue rectangles, primers used in methylation-specific PCR (MSP); in green rectangles, primers used in real-time PCR. (**B**) Graphical representation of MSP output: a glioblastoma sample (S1), an internal methylated (CT) and a non-methylated (CT-) controls, and a control without DNA (B) are depicted, each including a lane signed with «+» (methylated DNA specific primers) and a lane signed with «−» (un-methylated DNA specific primers). (**C**) Graph indicating *MGMT* methylation status predicted by methylome analysis. (**D**) Capillary electrophoresis indicating output of pyrosequencing analysis of *MGMT* promoter methylation in 2 glioblastoma samples.

**Figure 3 ijms-23-07148-f003:**
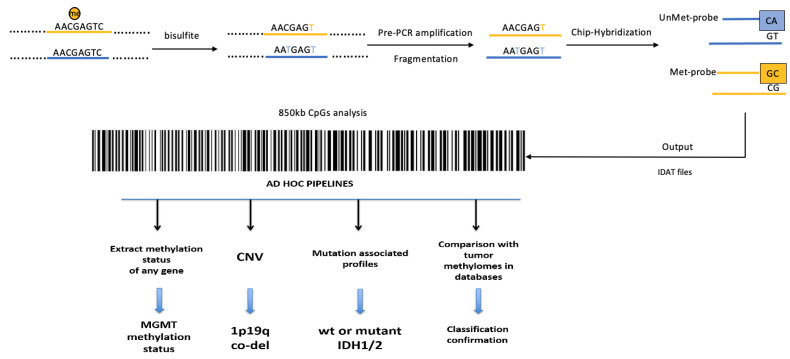
Methylome workflow and data analysis: schematic representation of the analytic steps to perform the EPIC array. DNA is converted with sodium bisulfite, pre-amplificated and fragmented. DNA is, then, hybridized on a specific array with specific probes recognizing the single CG both if methylated or if not methylated. The technology analyzes 850,000 CpG sites tracing a sort of tumor barcode. Output raw data are analyzed using different bioinformatic pipelines to extrapolate information useful for the characterization of the tumor, such as the methylation status of the *MGMT* gene, the copy number variations (CNVs), the co-deletion 1p-19q, the presence of functional mutations in the *Isocitrate Dehydrogenase (IDH) 1 and 2* genes. Using a specific comparison algorithm, the analyzed methylomic profile is compared with others present in the database for a more precise classification of the tumor.

**Figure 4 ijms-23-07148-f004:**
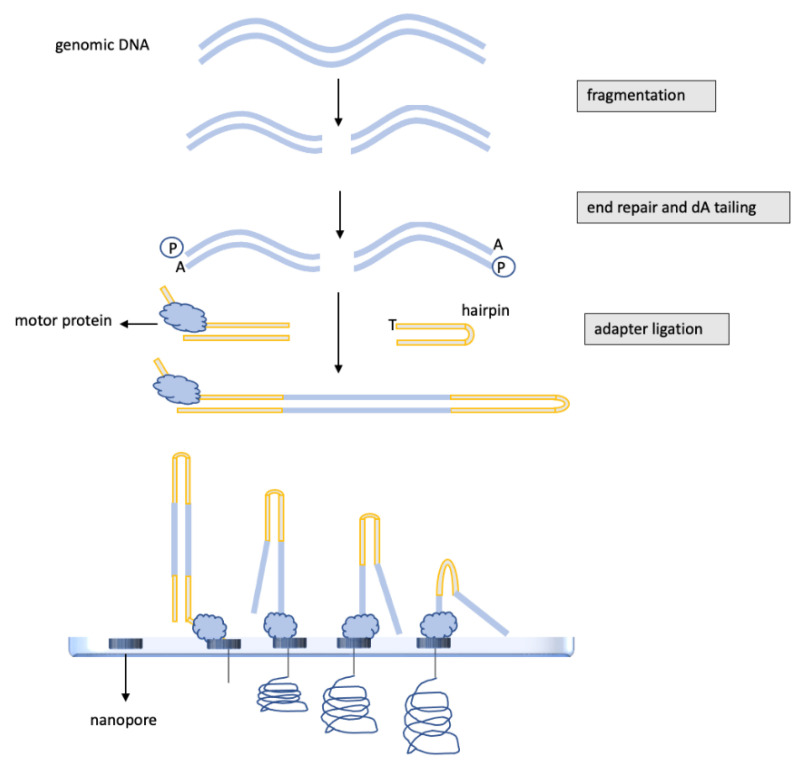
Nanopore sequencing strategy. As the first step, the genomic DNA is fragmented and then subjected to end repair and dA tailing. To allow DNA passage through flow cell pores, DNA is ligated to a protein adapter. Each flow cell may contain from 512 to 10,700 channels that may be potentially suitable for sequencing from 10–20 Gigabases to up to 100 Gigabases. Each channel is linked to a specific electrode, and a constant voltage is applied across the membrane. When double-stranded DNA molecules cross the pores, the voltage changes according to the specific single nucleotide passing the membrane. The difference in electric potential between the two sides of the membrane is converted into the specific nucleotide that may be identified in real-time during nanopore runs.

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
