# Peer review of "MGMT and Whole-Genome DNA Methylation Impacts on Diagnosis, Prognosis and Therapy of Glioblastoma Multiforme"

_ijms, 2022, doi:10.3390/ijms23137148_

Round 1

Reviewer 1 Report

The manuscript titled “DNA methylation impact on diagnosis, prognosis, and therapy of glioblastoma multiforme” by Della Monica, R.; et al. is a review where the authors deeply study the DNA methylation process in brain tumors by the O6-methylguanine-DNA-methyltransferase (MGMT) and how it influences on the regulation of the enzymes involved in glioblastoma multiforme cancer diseases. The impact of multiple factors is taken into account (MGMT, drugs as temozolomide, the action of DNA methyltransferase inhibitor 5-azacytidine, among others). The structure of the sections which conform the review is sequential and comprehensive for potential readers (1. Introduction; 2. Relevance of MGMT methylation assessment for glioblastoma clinical management; 3. Techniques for MGMT methylation testing in glioblastomas; 4. Whole-genome methylation profiling (methylome) of glioblastomas; 5. Glioblastoma DNA methylation assessment in liquid biopsies; 6. DNA methylation analysis of glioblastomas by nanopore, “third generation” sequencing; 7. IDH1/2 mutations and the methylator phenotype of glioblastomas: new therapeutic targets; and 8. Conclusions). The gathered findings may be relevant for the examined field. The results achieved are well-discussed during the main body of the reported manuscript. The scientific paper is well written. In my opinion the present manuscript is innovative and the methodological approached used matches with the scope of International Journal of Molecular Sciences. For the above described reasons, I recommend the publication in International Journal of Molecular Sciences once the following remarks will be fixed:

--------

TITLE

Authors should take care about “impact” verb conjugation. Please, change the current title by “DNA methylation impacts on diagnosis, prognosis, and therapy of glioblastoma multiforme”.

--------

REST OF THE SECTIONS

In general terms, the article are well-discussed and the cited reference are appropriate. The following aspects should be further discussed:

I)          Authors present temozolomide (from line 48) as efficient treatment against Glioblastoma multiforme, but the potential side effects of this drug are completely ignored. Please, add some statements to explain this point. Some articles have been published focusing on this point [1,2]. Moreover, it has been recently reported that other chemical agents as lithium [3] or nanoparticles decorated with bromodomain inhibitors [4] could positively enhance the antitumor effect of temozolomide against the studied cancer disease. In addition, it has been recently published some work indicating DNA polymerase can protect patient cells by the effect of temozolomide compound [5].

II)        I strongly encourage to add an extra Figure remarking the chemical reactions involved when termozolomide and GMT perform their action on DNA strands.

III)    The present review covers the diagnosis strategies of glioblastoma multiforme patients. What do the authors think about optical genome mapping (OGM) technique? OGM is a tool newly developed to identify gene mutations. It could be used as alternative diagnosis technique to find methylation processes [6]. Moreover, the present review presents pyrosequencing as quantitative technique to evaluate methylation effects made by MGMT. What do the authors thing about OGM to partially supply this task? In case affirmative, authors should highlight the potential and drawbacks comparing both techniques.

IV)    Authors should also add some further data concerning the statistical analysis of clinical trials regarding glioblastoma multiforme patients treated with temozolomide.

V)       Liquid biopsies section (from line 245) should add further quantitative information regarding if it exists any dilution effect of methylation process necessary to be taken into account.

VI)    Section 8. Conclusions (from line 372) should be modified to “8. Conclusions and future perspectives” by annexing some prospects. This fact will significantly aid to potential readers to better understand the importance of the present review.

[1] Bae, S.H.; et al. Toxicity profile of temozolomide in the treatment of 300 malignant glioma patients in Korea. J. Korean Med. Sci. 2014, 29, 980-984. https://doi.org/10.3346/jkms.2014.29.7.980.

[2] Kuter, D.J. Treatment of chemotherapy-induced thrombocytopenia in patients with non-hematologic malignancies. Haematologica. 2022, 107, 1243-1263. https://doi.org/10.3324/haematol.2021.279512.

[3] Han, S.; et al. Lithium enhances the antitumor effect of temozolomide against TP53 wild-type glioblastoma cells via NFAT1/FAsL signaling. Br. J. Cancer 2017, 116, 1302-1311. https://doi.org/10.1038/bjc.2017.89.

[4] Lam, F.C.; et al. Enhanced efficacy of combined temozolomide and bromodomain inhibitor therapy for gliomas using targeted nanoparticles. Nat. Commun. 2018, 9, 1991. https://doi.org/10.1038/s41467-018-04315-4.

[5] Latancia, M.T.; et al. DNA polymerase eta protects human cells against DNA damage induced by the tumor chemotherapeutic temozolomide. Mutat. Res. Genet. Toxicol. Environ. Mutagen. 2022, 878, 503498. https://doi.org/10.1016/j.mrgentox.2022.503498.

[6] Rusk, N. Optically mapping methylation. Nat. Methods 2019, 16, 362. https://doi.org/10.1038/s41592-019-0420-0.

--------

OVERVIEW AND FINAL COMMENTS

The submitted work is well-designed and the gathered results are interesting for the biomedical and clinical fields in special related to carcinogenic glioblastoma multiforme disorders. For this reason, I will recommend the present scientific manuscript for further publication in International Journal of Molecular Sciences once all the aforementioned suggestions will be properly fixed.

Author Response

Dear Reviewer,

please consider this thoroughly revised version of our manuscript entitled “MGMT and whole-genome DNA methylation impacts on diagnosis, prognosis and therapy of glioblastoma multiforme”, number IJMS-1771742, by Della Monica et al. Please note that, as suggested by you and the other Reviewer, the title has changed from the original “DNA methylation impact on diagnosis, prognosis and therapy of glioblastoma multiforme”.

We wish to thank you for the favourable evaluation of our manuscript and, mostly, for your punctual criticisms that significantly helped to improve it.

A point-to-point reply to your comments follows. For your convenience, all the changes have been tracked in the “Della Monica et al. revised” file.

Thank you again for the attention you have dedicated to our work.

Sincerely,

Roberta Visconti, on behalf of all authors.

TITLE

Authors should take care about “impact” verb conjugation. Please, change the current title by “DNA methylation impacts on diagnosis, prognosis, and therapy of glioblastoma multiforme”.

Our replay:

As suggested, the title has been modified also taking into account the other reviewer’s comments.

REST OF THE SECTIONS

In general terms, the article are well-discussed and the cited reference are appropriate. The following aspects should be further discussed: 

I) Authors present temozolomide (from line 48) as efficient treatment against Glioblastoma multiforme, but the potential side effects of this drug are completely ignored. Please, add some statements to explain this point. Some articles have been published focusing on this point [1,2]. Moreover, it has been recently reported that other chemical agents as lithium [3] or nanoparticles decorated with bromodomain inhibitors [4] could positively enhance the antitumor effect of temozolomide against the studied cancer disease. In addition, it has been recently published some work indicating DNA polymerase can protect patient cells by the effect of temozolomide compound [5]. 

Our replay:

As suggested, temozolomide side effects have been discussed (lines 60-63, references 5-6). Moreover, we have discussed the three relevant studies, you appropriately pointed out, investigating novel temozolomide combinatorial regimes (lines 64-70, references 7-8-9).

II) I strongly encourage to add an extra Figure remarking the chemical reactions involved when termozolomide and GMT perform their action on DNA strands.

Our replay:

As suggested, we have added an extra figure (Figure 1) depicting temozolomide and MGMT effects on DNA.

III) The present review covers the diagnosis strategies of glioblastoma multiforme patients. What do the authors think about optical genome mapping (OGM) technique? OGM is a tool newly developed to identify gene mutations. It could be used as alternative diagnosis technique to find methylation processes [6]. Moreover, the present review presents pyrosequencing as quantitative technique to evaluate methylation effects made by MGMT. What do the authors thing about OGM to partially supply this task? In case affirmative, authors should highlight the potential and drawbacks comparing both techniques.

Our replay:

As suggested, we have discussed OGM and its potential as a quantitative technique for investigating DNA methylation in glioblastoma (lines 483-497, references 80-81).

IV) Authors should also add some further data concerning the statistical analysis of clinical trials regarding glioblastoma multiforme patients treated with temozolomide.

Our replay:

As suggested, some further data concerning the statistical analysis of clinical trials have been added (lines 105-109; lines 117-118; lines 124-126; lines 128-129).

V) Liquid biopsies section (from line 245) should add further quantitative information regarding if it exists any dilution effect of methylation process necessary to be taken into account.

Our replay:

As suggested, this point has been discussed (lines 401-403).

VI) Section 8. Conclusions (from line 372) should be modified to “8. Conclusions and future perspectives” by annexing some prospects. This fact will significantly aid to potential readers to better understand the importance of the present review.

Our replay:

As suggested, the title of section (section number 6 in the revised manuscript) has been modified and we have add some more prospects (lines 474-476; lines 490-497).

[1] Bae, S.H.; et al. Toxicity profile of temozolomide in the treatment of 300 malignant glioma patients in Korea. J. Korean Med. Sci201429, 980-984. https://doi.org/10.3346/jkms.2014.29.7.980.

[2] Kuter, D.J. Treatment of chemotherapy-induced thrombocytopenia in patients with non-hematologic malignancies. Haematologica2022107, 1243-1263. https://doi.org/10.3324/haematol.2021.279512.

[3] Han, S.; et al. Lithium enhances the antitumor effect of temozolomide against TP53 wild-type glioblastoma cells via NFAT1/FAsL signaling. Br. J. Cancer 2017116, 1302-1311. https://doi.org/10.1038/bjc.2017.89.

[4] Lam, F.C.; et al. Enhanced efficacy of combined temozolomide and bromodomain inhibitor therapy for gliomas using targeted nanoparticles. Nat. Commun20189, 1991. https://doi.org/10.1038/s41467-018-04315-4.

[5] Latancia, M.T.; et al. DNA polymerase eta protects human cells against DNA damage induced by the tumor chemotherapeutic temozolomide. Mutat. Res. Genet. Toxicol. Environ. Mutagen2022, 878, 503498. https://doi.org/10.1016/j.mrgentox.2022.503498.

[6] Rusk, N. Optically mapping methylation. Nat. Methods 201916, 362. https://doi.org/10.1038/s41592-019-0420-0.

OVERVIEW AND FINAL COMMENTS

The submitted work is well-designed and the gathered results are interesting for the biomedical and clinical fields in special related to carcinogenic glioblastoma multiforme disorders. For this reason, I will recommend the present scientific manuscript for further publication in International Journal of Molecular Sciences once all the aforementioned suggestions will be properly fixed.

Reviewer 2 Report

The authors summarized the significance of MGMT promoter methylation, the effects of IDH mutation in gliomas, and the current techniques for detecting methylation in gliomas. This review focuses on the detection methods for single-gene or whole-genome methylation. Now only the part of MGMT methylation is closely linked to the title. So it would be helpful if the title could be changed to be more consistent with the content of this review.

The organization of the review could be improved. The part of the technique description could be put together.

IDH mutation is present in  80% of Grade II or III glioma and secondary GBM.  It would be helpful if the authors fully discuss the effects of IDH mutation on DNA methylation in gliomas. 

Author Response

Dear Reviewer,

please consider this thoroughly revised version of our manuscript entitled “MGMT and whole-genome DNA methylation impacts on diagnosis, prognosis and therapy of glioblastoma multiforme”, number IJMS-1771742, by Della Monica et al. Please note that, as suggested by you and the other Reviewer, the manuscript title has changed from the original “DNA methylation impact on diagnosis, prognosis and therapy of glioblastoma multiforme”.

We wish to thank you for the favourable evaluation of our manuscript and, mostly, for your punctual criticisms that significantly helped to improve it.

A point-to-point reply to your comments follows. For your convenience, all the changes have been tracked in the “Della Monica et al. revised” file.

Thank you again for the attention you have dedicated to our work.

Sincerely,

Roberta Visconti, on behalf of all authors.

The authors summarized the significance of MGMT promoter methylation, the effects of IDH mutation in gliomas, and the current techniques for detecting methylation in gliomas. This review focuses on the detection methods for single-gene or whole-genome methylation. Now only the part of MGMT methylation is closely linked to the title. So it would be helpful if the title could be changed to be more consistent with the content of this review.

Our replay:

As suggested, the title has been modified to be more consistent with the content of the review.

The organization of the review could be improved. The part of the technique description could be put together.

Our replay:

As suggested, we have described the techniques for assessing DNA methylation in one paragraph (#3: Techniques for DNA methylation testing in glioblastomas), divided in three subparagraphs (#3.1: Techniques for MGMT methylation assessment in glioblastomas; #3.2: Whole-genome methylation profiling (methylome) of glioblastomas; #3.3: DNA methylation analysis of glioblastomas by nanopore, “third-generation”, sequencing).

 IDH mutation is present in 80% of Grade II or III glioma and secondary GBM.  It would be helpful if the authors fully discuss the effects of IDH mutation on DNA methylation in gliomas.

Our replay:

As suggested, we have discussed in more detail the role of the IDH1/2 mutations in gliomas, including their effects on DNA methylation (lines 415-421; lines 428-436; newly added references 66-67-70).

Round 2

Reviewer 2 Report

The authors addressed my major concerns by adding more detailed discussions and reorganization. The revised version of the manuscript appears to be good. It looks ready for publication as far as I can tell.